**Data Availability Statement:** All relevant data are within the manuscript and its Supporting Information files.

**Funding:** WW is in receipt of an award funding the work from Action for A-T (https://actionforat.org/),

# The natural history of ataxia-telangiectasia (A-T): A systematic review

Emily Petley[1], Alexander Yule[2], Shaun Alexander[1], Shalini Ojha[1,3]*, William P. Whitehouse[1,4]

1 School of Medicine, University of Nottingham, Nottingham, United Kingdom, 2 United Lincolnshire Hospitals NHS Trust, Lincoln, United Kingdom, 3 Children's Hospital, University Hospitals of Derby and Burton, NHS Foundation Trust, Derby, United Kingdom, 4 Nottingham Children's Hospital, Nottingham University Hospital NHS Trust, Nottingham, United Kingdom

* shalini.ojha@nottingham.ac.uk

## Abstract

### Background

Ataxia-telangiectasia is an autosomal recessive, multi-system, and life-shortening disease caused by mutations in the ataxia-telangiectasia mutated gene. Although widely reported, there are no studies that give a comprehensive picture of this intriguing condition.

### Objectives

Understand the natural history of ataxia-telangiectasia (A-T), as reported in scientific literature.

### Search methods

107 search terms were identified and divided into 17 searches. Each search was performed in PubMed, Ovid SP (MEDLINE) 1946-present, OVID EMBASE 1980 –present, Web of Science core collection, Elsevier Scopus, and Cochrane Library.

### Selection criteria

All human studies that report any aspect of A-T.

### Data collection and analysis

Search results were de-duplicated, data extracted (including author, publication year, country of origin, study design, population, participant characteristics, and clinical features). Quality of case-control and cohort studies was assessed by the Newcastle-Ottawa tool. Findings are reported descriptively and where possible data collated to report median (interquartile range, range) of outcomes of interest.

### Main results

1314 cases reported 2134 presenting symptoms. The most common presenting symptom was abnormal gait (1160 cases; 188 studies) followed by recurrent infections in classical

A-T Society (https://www.atsociety.org.uk/) and BrAshA-T (https://brashat.org.au/). The funders had no role in study design, data collection and analysis, decision to publish, or preparation of the manuscript.

**Competing interests:** The authors have declared that no competing interests exist.

ataxia-telangiectasia and movement disorders in variant ataxia-telangiectasia. 687 cases reported 752 causes of death among which malignancy was the most frequently reported cause. Median (IQR, range) age of death (n = 294) was 14 years 0 months (10 years 0 months to 23 years 3 months, 1 year 3 months to 76 years 0 months).

## Conclusions

This review demonstrates the multi-system involvement in A-T, confirms that neurological symptoms are the most frequent presenting features in classical A-T but variants have diverse manifestations. We found that most individuals with A-T have life limited to teenage or early adulthood. Predominance of case reports, and case series demonstrate the lack of robust evidence to determine the natural history of A-T. We recommend population-based studies to fill this evidence gap.

## Introduction

Ataxia-telangiectasia (A-T) is an autosomal recessive, multi-system, progressive and life-shortening disease due to mutations in the ataxia-telangiectasia mutated (ATM) gene on chromosome 11q.26. The severest form, classical A-T, most often caused by a truncating mutation, results in either the absence of ATM protein or its ATM kinase activity. Variant form with reduced kinase activity presents with a milder phenotype and a slower disease progression [1].

A-T generally presents at 12–18 months with an unsteadiness of gait due to cerebellar ataxia. The ataxia gradually worsens and by the age of 10 years children are unable to walk. Other features such as dysarthria, oculomotor apraxia, dysphagia, choreoathetosis, dystonia, tremor, myoclonus, and peripheral neuropathy gradually develop and often worsen. The majority do not have severe cognitive impairment in childhood, although progressive cognitive impairment has been reported over time [2, 3]. Telangiectasia, the other eponymous feature, develops at 3–4 years of age, mostly in the bulbar conjunctiva but can sometimes be found in other organs such as the bladder. Immunological deficits make individuals with A-T more prone to recurrent infections, particularly sinopulmonary infections with progressive deterioration of lung function. Increased risk of malignancies such as leukaemia, lymphoma, and solid tumours further impact longevity with life expectancy generally limited to 20–30 years of age in people with classical A-T.

This wide spectrum of manifestations and multi-disciplinary interest in A-T means that numerous academic papers have been published on this condition. Whilst textbook and narrative reviews exist [4], no attempt has ever been made to collate the available information to give a complete, multi-faceted picture of this intriguing condition. The aim of this study is to perform a systematic review of all scientific literature reporting the natural history of A-T.

### Aims and objectives

To describe the natural history of ataxia-telangiectasia (A-T) from birth to death as presented in existing scientific literature.

P–People of all ages, gender and ethnicity
I (E)–Diagnosis of ataxia-telangiectasia
C–People without ataxia-telangiectasia (where comparison group included)
O—Age of onset of cerebellar gait ataxia

Age of wheelchair use
Length of survival and cause of death
Presenting features of A-T
Understanding levels of AFP throughout life course of A-T

# Methods

## Protocol and registration

The review protocol can be accessed at Open Science Framework [5].

## Eligibility criteria

All study types were included. There were no restrictions on length of follow-up, or type of publication.

## Information sources

Six databases (PubMed; Ovid SP (MEDLINE) 1946- present; OVID EMBASE 1980 –present; Web of Science core collection; Elsevier Scopus (Categories; medicine, biochemistry, genetics and molecular biology, immunology and microbiology, neuroscience, pharmacology, toxicology and pharmaceutics, health professions); and the Cochrane Library) were searched from the date of the database creation to 19[th] August 2021.

## Search

Initially A-T was identified by combining "Ataxia-telangiectasia"; "Ataxia-telangectasia"; "Ataxia telangiectasia" "Ataxia telangectasia"; "Louis-Bar"; and "Louis Bar" with the 'OR' function. A further 103 search terms were grouped into 17 searches and then combined with the above search using the 'AND' function. The full search strategy is given in S1 Protocol. In order to ensure that no relevant search terms were missed both UK and US English spellings were included, truncating was used where appropriate and common misspellings, for example 'telangectasia' were included.

## Study selection

Included studies were selected as described in Table 1.

The review includes reports of cases of A-T at all ages (children and adults). Cases of classical and variant A-T were included. Cases were identified as variant A-T if reported as such or reported to have some ATM protein kinase activity. Other participants were presumed to have classical A-T.

**Table 1. Criteria for study selection for review of natural history of ataxia-telangiectasia.**

|  | Inclusion criteria | Exclusion criteria |
|---|---|---|
| **Participants** | All ages and gender with a diagnosis of A-T | Animals, plants, or no cases with a diagnosis of A-T |
| **Type of article** | Original research articles/data | Review articles, not original articles |
| **Clinical relevance** | Described clinical data | Laboratory or animal data only |
| **Location** | All countries | N/A |

A-T, ataxia-telangiectasia; N/A, not applicable

### Data collection process

All titles and articles were downloaded to a citation software (Endnote X9; Clarivate Analytics, Philadelphia) and duplicates removed automatically. The search was uploaded into a review software (Covidence systematic review software, Veritas Health Innovation, Melbourne, Australia. Available at www.covidence.org) which identified and removed some more duplicates. The remaining articles were sorted by title, year, journal, and authors, and remaining duplicates were manually removed.

One author (EP) screened all titles and abstracts and selected the full text articles. Full text articles were reviewed by EP who extracted data using a bespoke data extraction form (Microsoft Excel, 2016 Microsoft Corporation, United States). Any data extraction difficulties were discussed and resolved with two authors (SO and WW).

The extracted data included author, year of publication, country of origin, study design, study population, number of cases of A-T in study' participant characteristics such as age, gender, clinical features related to the review's primary and secondary outcomes.

No assumptions were made during data collection. Only statements about the presence or lack of presence of an outcome were included in the analysis.

Where reported, age of onset/diagnosis for each outcome was extracted. Where symptoms were reported as having occurred 'by' an age and the age of onset was not determinable, it was not included.

## Outcomes

### Primary outcomes

- Age of onset of cerebellar gait ataxia

- Age of wheelchair use

- Length of survival and cause of death

- Presenting features of A-T

- Understanding levels of AFP throughout life course of A-T

### Secondary outcomes

- Missed and incomplete diagnoses

- Reasons for diagnostic delays

- Age of onset of other neurological signs and symptoms, for example movement disorders, dysarthria, developmental delay, imaging findings

- Other diagnosis, types, age of onset and treatments (where available)

  - Common recurrent infections

  - Respiratory conditions including bronchiectasis, interstitial lung disease

  - Malignancies

  - Diabetes

  - Granulomatous disease

  - Skin conditions

- Use of gastrostomy (reasons and age of insertion)

- Laboratory findings including vitamin D, dyslipidaemia

- Any other findings

### Assessment of risk of bias

Quality assessment of cohort and case-control studies was completed by EP and AY using the Newcastle-Ottawa tool [6], as recommended by the Cochrane Collaboration [7]. The ratings for cohort studies were converted to ARHQ standards [8].

### Identification of multiple reports of same cases

In addition to removing duplicates, we identified and combined multiple reports of the same cases, where identifiable and possible. Initial full text review revealed that some cases were included in several reports. We identified such duplications by pattern recognition and matching them on characteristics such as age and gender of the case, presence of unusual diagnoses or other common features, authors, and site of study. The information from such reports were then combined such that in the analyses they represented one patient. However, we acknowledge that not all multiple reports of the same individual can be identified in this manner. Where we were unable to reasonably ascertain that the reports were of the same case, we included them as individual cases.

### Statistical analyses

The extracted data were analysed using calculations of total number of a sign/symptom/diagnosis, age range, and median age of onset or diagnosis (dependent on variable). Findings are reported descriptively and where possible data are collated to report median (range, interquartile range) of each presentation or feature of the condition. Statistical analysis was performed in Microsoft Excel 2016 (Microsoft, Redmond).

### Dealing with missing data

This review is limited to the data that were available in the included studies. Due to the large number of studies and large volume of missing data, it was not feasible to contact the authors to attempt full data collection on each included case.

### Subgroup analysis

A sub-group analysis was performed with the same method as above of cases with presumed or confirmed variant A-T and those with presumed or confirmed classical A-T.

The PRISMA check list was used in compiling this report, S1 Checklist.

## Results

### Results of the search

The search yielded 209086 titles and abstracts (Fig 1). After removal of 193404 duplicates and exclusion of 14399 articles by review of title and abstract, 1283 full text articles were reviewed.

### Included studies

We included 1131 studies of eight different types: 434 case reports, 378 case series, 100 cross-sectional, 70 case-control, 57 cohort, 60 prevalence, 29 interventional, and 3 qualitative studies.

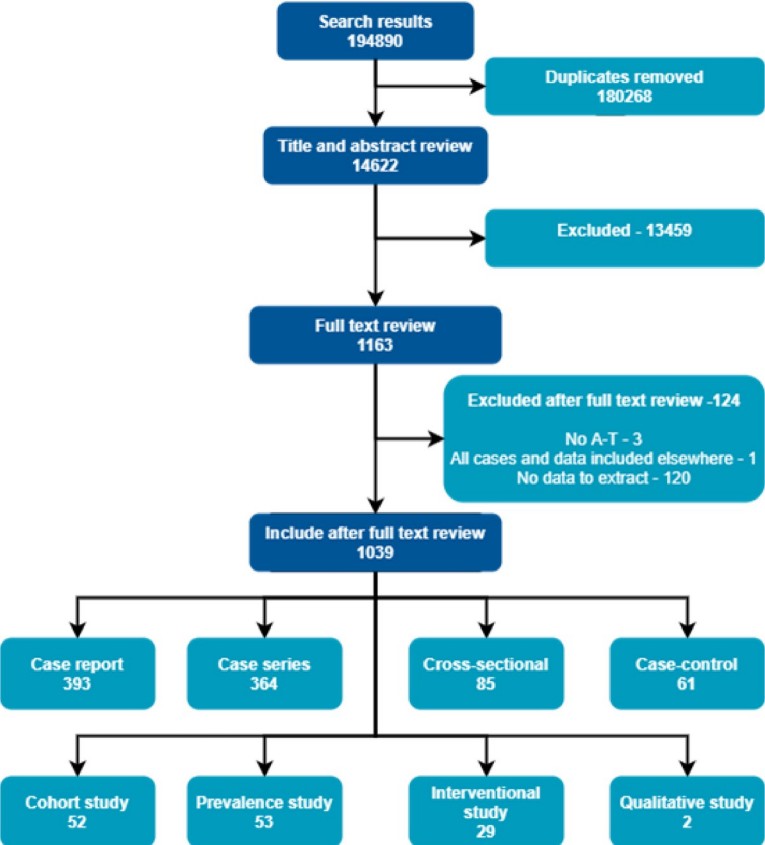

**Fig 1. PRISMA diagram.**

Most studies included fewer than 10 cases although there were 33 studies with more than 100 cases each (Fig 2). The median (IQR, range) number of participants per study was 2 (1 to 12, 1 to 585). Six studies [9–14] did not report the number of participants.

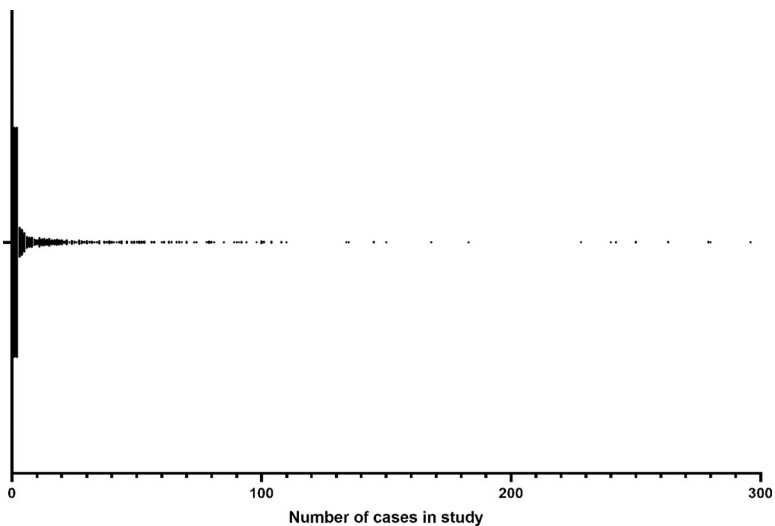

**Fig 2. Number of cases per study.**

A total of 18247 participants were included in these studies. Median age at inclusion was (IQR) (n = 1648) 144.0 months (84.0–240.0). The youngest case was of a 6 day old infant diagnosed by newborn screening programme and the oldest was 78 years of age. Sex was reported in 7840 cases of which 3719 (47.4%) were female. There were 457 (2.5%) confirmed/presumed variant cases included in 60 reports [1, 15–73].

Studies were widely reported across North America, Europe, and parts of Asia. There were fewer reports from Africa, parts of South America and the Middle East.

## Family history

Of the 18246 cases, family history of A-T was reported in 1274 cases (Table 2) and 142 cases (53 studies) had 199 illnesses or symptoms other than A-T in a relative (Fig 3A). 1279 cases (109 studies [15, 53, 65, 71, 72, 78, 87, 88, 95, 99–101, 109, 110, 115, 118–221]) were the children of consanguineous relationships, and 186 cases (86 studies [23, 33, 41, 50, 54, 62, 63, 65, 71, 74, 84, 88, 95, 111, 116, 142, 146, 151–153, 188, 208–210, 212, 214, 216, 218, 219, 221–277]) were reported as being born of non-consanguineous relationships.

## Birth and early childhood

Gestational age at birth was reported in 320 cases (68 studies); 289 cases at term gestation, 31 cases <37 weeks gestation. The lowest gestation was reported as "< 30 weeks". Birth weight was reported in 41 cases (34 studies [43, 59, 63, 84, 108, 110, 111, 139, 178, 206, 212, 231, 242, 252, 254, 263, 268, 272, 276, 298, 305, 325, 372, 380, 384, 391, 399–406]) with median (range) of 2.9.5 (1.32 to 4.08) kg.

Antenatal problems were reported in 20 cases (12 studies [84, 106, 108, 129, 143, 231, 236, 263, 372, 384, 404, 407]) while 25 postnatal concerns were reported in 22 cases (12 studies [84, 214]).

Details are provided in S3-S5 Tables in S1 File.

## Diagnosis

329 cases reported an age of diagnosis as shown in Table 3.

17 cases (10 studies [106, 119, 124, 133, 204, 247, 261, 310, 359, 404]) reported a delay in diagnosis. Case reports were excluded from this analysis. Most cases were reported as being diagnosed at the first presentation. Most reported cases were diagnosed without any delay, however a minority were diagnosed late: the median delay in diagnosis (n = 17) was 0.0 i.e.,

**Table 2. Family history of ataxia-telangiectasia (A-T) in reported case of A-T.**

| Relation with A-T | Number of cases (number of studies) references |
|---|---|
| First degree relative | 710 (151) |
| | [17, 24, 27, 28, 33, 37, 41, 49, 58, 66, 72, 76, 90–92, 108, 109, 118–120, 129, 132, 133, 137–140, 145, 152, 161, 168, 170, 172, 175, 176, 180, 183, 185, 195, 196, 200, 204, 211, 221, 225–227, 235, 240, 245, 247–250, 256, 257, 261, 262, 265, 275, 278–369] |
| Second degree relative | 18 (5) |
| | [66, 109, 111, 161, 175, 227] |
| Third degree relative | 24 (6) |
| | [66, 124, 159, 206, 315, 370] |
| Unspecified relation | 522 (25) |
| | [44, 87, 108, 142–144, 154, 156, 162, 175, 177, 182, 195, 199, 211, 213, 352, 371–378] |

Absence of family history of A-T was documented in at least 60 cases (54 studies [21, 50, 65, 80, 86, 95, 98, 111, 122, 134, 135, 150, 155, 181, 188, 190, 197, 202, 203, 222, 224, 236, 238, 239, 242, 252, 254, 255, 267, 272–274, 276, 366, 379–398]).

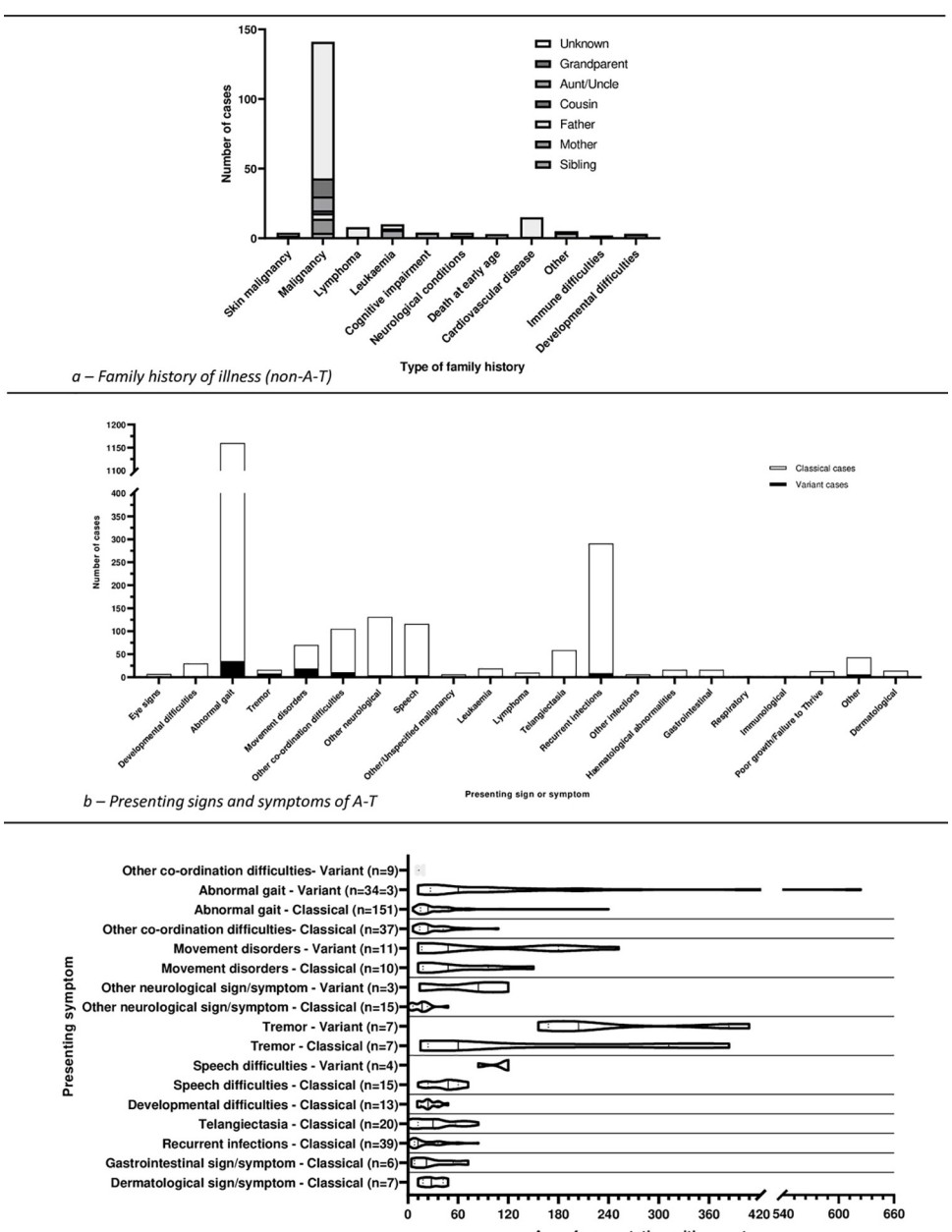

**Fig 3. Family history of other illness, and presenting symptoms and signs.**

diagnosed at first presentation but there was wide variation with a range of 0.0–312.0 (IQR, 0.0–43.0) months.

Missed, or incorrect diagnoses reported are shown in S1 Fig. Cerebral palsy was the most common incorrect diagnosis. 7 of the 14 cases reported with a specific type of cerebral palsy, had the ataxic form.

## Clinical features

The presenting sign(s)/symptom(s) were reported in 1314 cases. These included 2134 signs/symptoms (Fig 3B and 3C).

**Table 3. Age of diagnosis of ataxia-telangiectasia as reported in literature.**

| | Number of studies (references) | Number of cases | Median (IQR, range) (in months) |
|---|---|---|---|
| All cases | [221] | 329 | 72.0 (36.0–120.0, 0.7–720.0) |
| | [16, 18, 27, 30, 42, 47, 50, 54, 59, 62, 70, 74, 76, 80–83, 89, 99, 101, 103, 105, 106, 110, 111, 113, 116, 119, 121, 122, 124–126, 128, 134, 135, 140, 148, 150, 153, 154, 160, 164–166, 169, 174, 176, 177, 181, 184, 187, 188, 193, 196, 203, 205–208, 211, 214, 219, 221–223, 225, 226, 228–230, 236–238, 240, 241, 245–247, 252, 255, 257, 259, 261–264, 271, 272, 276, 281, 290, 302, 304, 310, 340, 354, 355, 359, 362, 365, 373, 375, 378, 382, 385, 386, 388, 390–392, 394, 396, 404, 405, 408–506] | | |
| Variant cases only | [14] | 14 | 354.0 (231.0–456.0, 24.0–720.0) |
| | [16, 18, 27, 30, 42, 47, 50, 54, 59, 62, 65, 70, 71, 363] | | |
| Classical cases only | [209] | 315 | 72.0 (36.0–108.0, 0.7–528.0) |
| | [27, 30, 42, 47, 50, 59, 62, 65, 70, 74, 76, 80–83, 89, 99, 101, 103, 105, 106, 110, 111, 113, 116, 119, 121, 122, 124–126, 128, 134, 135, 140, 148, 150, 153, 154, 160, 164–166, 169, 174, 176, 177, 181, 184, 187, 188, 193, 196, 203, 205–208, 211, 214, 215, 217, 219, 221–223, 225, 226, 228–230, 236–238, 240, 241, 245–247, 252, 255, 257, 259, 261–264, 271, 272, 276, 281, 290, 302, 304, 310, 340, 354, 355, 359, 362, 365, 373, 375, 378, 382, 385, 386, 388, 390–392, 394, 396, 404, 405, 408–508] | | |

18 studies [24, 87, 162, 164, 174, 182, 187, 188, 195, 204, 315, 509–515] reported the mean in a further 688 presumed/ confirmed classical cases. The mean age of diagnosis in this group (n = 1003) was 75.8 months.

## Neurological

**Ataxia and mobility.** Cerebellar gait ataxia was reported in 3223 cases, truncal ataxia in 357 cases and limb ataxia in 163 cases (Fig 4A).

3 cases (1 study [36]) reported ataxia at 12 months that no longer had ataxia at 48 months, 72 months, and 72 months respectively.

Fig 4B shows all reported age data for cerebellar gait ataxia, truncal ataxia, limb ataxia and mobility.

**Eye signs.** Data was reported within the included studies on oculomotor apraxia, strabismus, pursuit, nystagmus, and saccades (Fig 4C and 4D).

17 further cases (1 study [537]) may also have had strabismus (reported as lateral gaze deviation or squint).

**Other neurological features.** Within the included articles, data were reported on sensory examination, peripheral neuropathy, seizures, drooling, muscle atrophy, and contractures (Fig 4E and 4F).

**Tone, weakness and reflexes.** Included studies reported data on reflexes, muscle tone, and muscle weakness (Fig 5A).

Several cases had progression of the reflexes from normal to hyporeflexia over time.

**Dysarthria.** 1219 cases (177 studies [18, 28, 31, 33, 36, 38, 41, 42, 47–49, 52, 57, 59, 62, 63, 65, 66, 72, 73, 75, 76, 80, 84, 87, 95, 99, 100, 103, 106, 108, 109, 111, 116, 118–120, 122, 123, 126, 129, 131, 138–141, 143, 147, 159, 165, 166, 174–176, 179, 181, 190, 191, 202, 206, 208, 211, 213, 216, 217, 219–226, 228–231, 233, 234, 236–238, 240, 242, 245, 247, 248, 251, 254, 255, 260, 265, 268, 271, 276, 278, 285, 287, 288, 290, 298, 303, 305, 310, 319, 323, 325, 326, 331, 335, 339, 342, 345, 347, 348, 363, 366, 368, 369, 372, 379, 380, 384, 388–390, 392, 394–396, 399, 401, 404, 405, 407, 409–411, 414, 415, 431, 435, 439, 440, 448, 449, 469, 472, 476, 479, 490, 494, 495, 501, 518, 519, 524, 528, 529, 534, 537, 538, 540, 541, 544, 555, 563, 574, 585, 586, 590, 604, 607, 609,

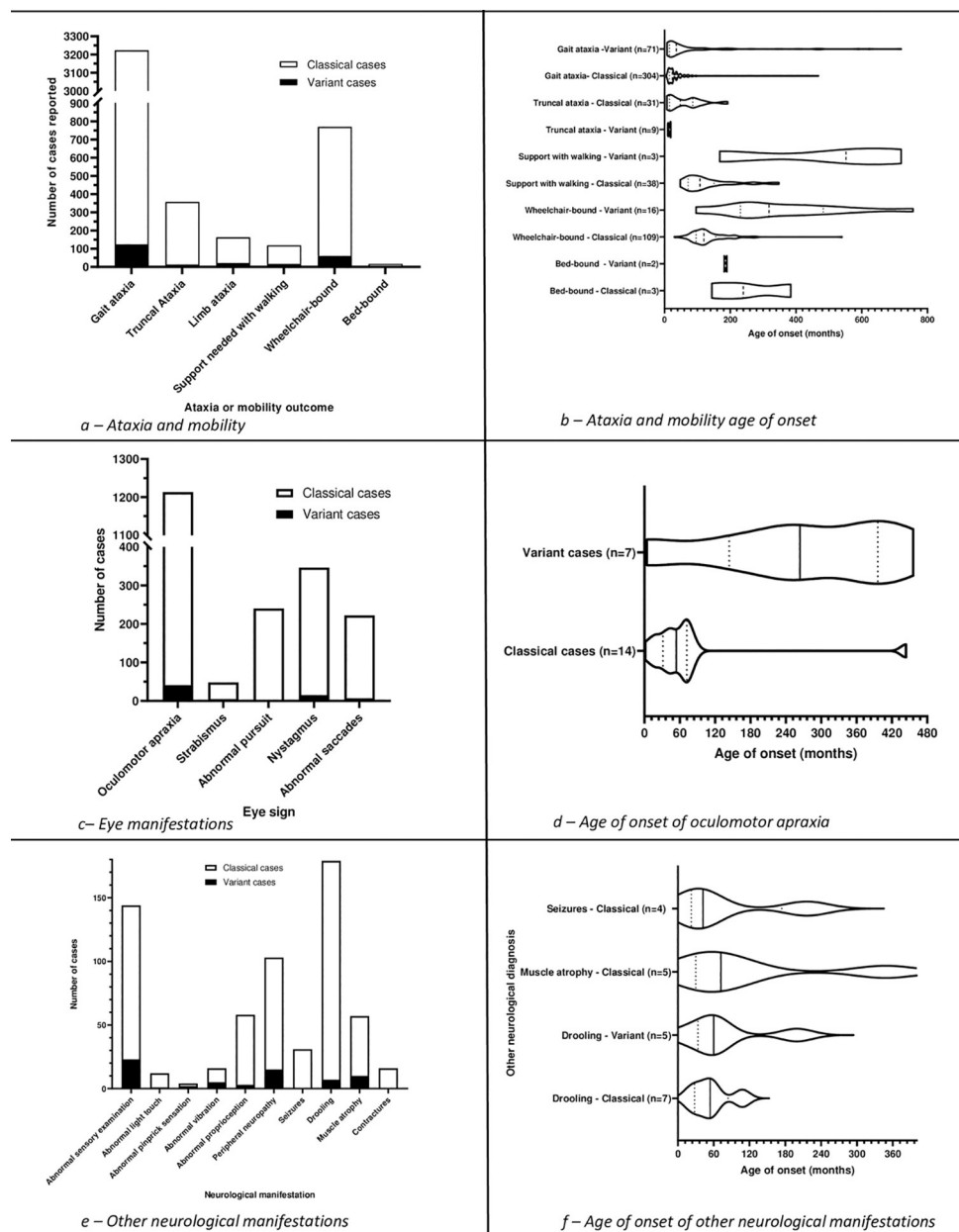

**Fig 4. Ataxia, mobility, eye movements, oculomotor apraxia, and other neurological manifestations.**

620, 625, 629, 633, 651]) reported dysarthria, 39 presumed/confirmed variant cases and 1180 in presumed/confirmed classical cases. Overall, the median age of onset (n = 58) was 60 months (range 12.0–528.0 months, IQR 36.0–96.0 months).

**Movement disorders.** Included studies reported a wide range of movement disorders (Fig 5B and 5C).

Data were reported on sites of dystonia; 6 cases, upper limb; 7 cases, cervical; 2 cases retrocollis; 2 cases laryngeal; 2 cases truncal; 5 cases, cervical, trunk and limb dystonia; 1 case, leg; 1 case, head; 1 case, oromandibular; and 1 case, finger dystonia.

**Cerebellar signs.** 107 included studies reported cerebellar signs (Fig 5D).

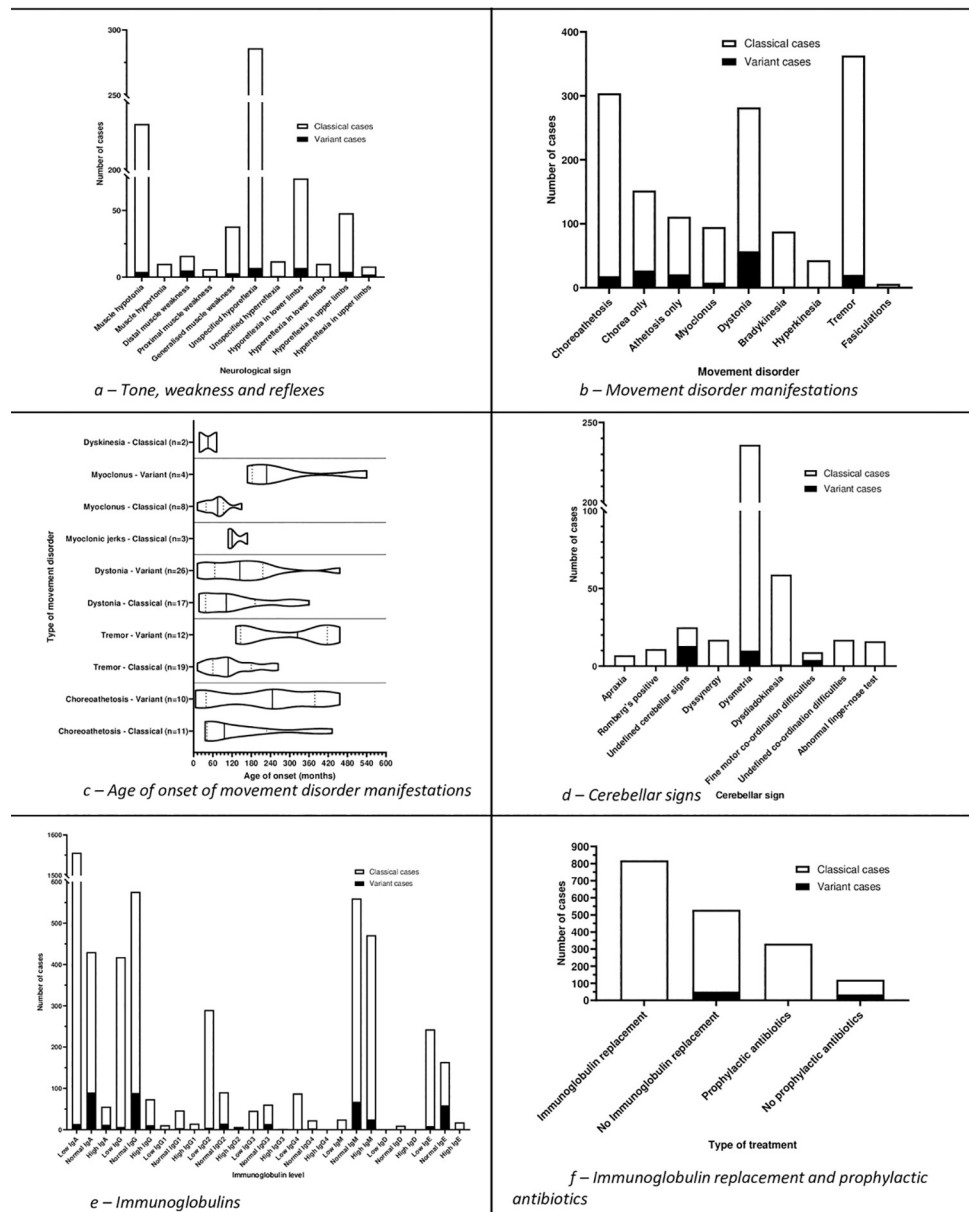

**Fig 5. Tone and weakness, movement disorders, cerebellar signs, immunoglobulin levels, immunoglobulin replacement, and prophylactic antibiotics.**

## Neuroimaging findings

546 cases (156 studies) reported abnormal neuroimaging (MRI or CT). Cerebellar atrophy/hypoplasia was the most common neuroimaging finding (Fig 8C).

All cerebellar atrophy was reported on MRI, except for 46 cases; 21 cases (8 studies [119, 123, 139, 145, 159, 180, 299, 392]) reported it after CT scan, 12 cases (7 studies [108, 231, 276, 345, 347, 358, 464]) reported it at post-mortem, 11 cases (5 studies [191, 225, 355, 378, 467]) reported cerebellar atrophy but did not report the imaging modality, and 2 cases [237, 298] reported it on pneumoencephalogram.

**Electromyography (EMG).** 62 cases (27 studies [33, 38, 41, 49, 53, 55, 72, 118, 120, 123, 139, 143, 186, 208, 212, 213, 338, 358, 366, 368, 372, 388, 392, 519, 528, 683, 684]) reported an abnormal EMG. The youngest age at which an abnormal EMG was reported was 4 years 0 months. The oldest age a normal EMG reported was 18 years 0 months. 16 cases were reported to have both abnormal motor and sensory nerve conduction. 1 case was reported to have only abnormal motor nerve conduction, and 10 cases were reported to only have abnormal sensory nerve conduction.

## Immunology

**Immunoglobulin levels and replacement.** Reported immunoglobulin levels are shown in (Fig 5E).

819 cases (147 studies) reported the use of immunoglobulin replacement therapy (Figs 5F and 6A). 3 cases (1 study [24]) were received immunoglobulin replacement temporarily. 2 variant cases were reported to receive immunoglobulin replacement [66].

**Prophylactic antibiotics.** 332 cases (56 studies) reported the start of use of prophylactic antibiotics (Figs 7F and 8A) including one [243] who had prophylactic antibiotics post-splenectomy.

## Recurrent infections

1326 cases reported recurrent infections (Fig 6B).

Further breakdown of recurrent infections is available in S2 Fig.

## Non-infectious respiratory manifestations

Studies included in the review reported non-infectious manifestations including bronchiectasis, chronic lung disease, pneumothorax, asthma, allergic rhinitis, bronchitis, pneumonitis, and obstructive sleep apnoea (Fig 6C and 6D).

259 cases reported bronchiectasis. The youngest age at which bronchiectasis was diagnosed was < 3 years [228]. The oldest child reported with no bronchiectasis was 108.0 months (n = 3) [43] and was in the presumed/confirmed variant group.

50 cases (13 studies) reported pneumothorax. 2 cases (2 studies [91, 240]) reported bilateral pneumothoraces. A further 5 cases (1 study [175]) reported that they had 2 pneumothoraces, but it could not be discerned if it was bilateral or two separate events. 2 cases (1 study [712]) were after gastrostomy tube insertion.

## Malignancy

1889 malignancies were reported in 1706 cases (365 studies). Only malignant tumours were included (Fig 6E and 6F).

The median age of diagnosis of NHL (n = 85) reported was 116.4 months (range 6–427.2 months, IQR 72.0–168.0 months). The median age of diagnosis of Hodgkin's disease (n = 61) reported was 108.0 months (range 44.0–684.0 months, IQR 96.0–166.0 months). The median age of diagnosis of leukaemia (n = 99) reported was 132.0 months (range 1.0–612.0 months, IQR 54.0–204.0 months).

Further breakdown of the results is available in the supplementary files, including the presenting symptoms of Hodgkin's lymphoma, non-Hodgkin's lymphoma and leukaemia (S3–S5 Figs).

## Alpha-feto protein (AFP) levels

1685 cases (292 studies [21, 22, 24, 27, 28, 30, 33, 35–38, 41, 42, 47–50, 52, 54, 74, 78, 83, 88, 89, 91, 99–101, 103, 105–107, 109, 115, 119–124, 126, 128, 129, 131–135, 138, 140, 143, 147, 148,

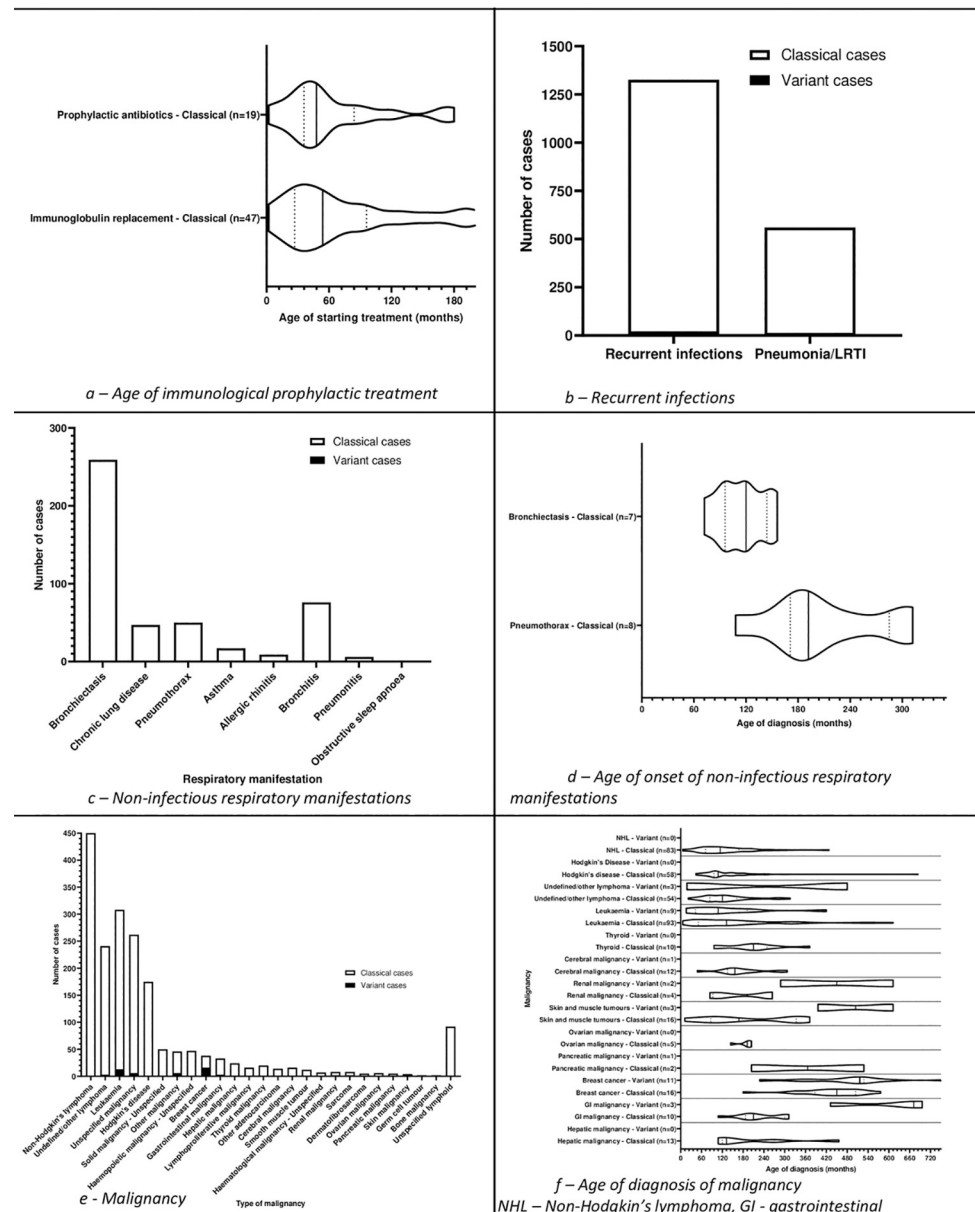

**Fig 6. Age at start of prophylactic antibiotic and immunoglobulin replacement, non-infectious respiratory manifestations, and malignancy.**

150, 160, 161, 165, 166, 168, 169, 172, 174, 177, 180, 183, 184, 191–193, 195–197, 199, 214, 223, 225, 227–229, 233, 236, 238, 239, 241, 244–248, 250–252, 256–259, 261, 263–270, 272, 273, 287, 294, 299, 300, 303, 304, 307, 312, 316, 318, 326, 338, 346, 354, 355, 359, 378, 381–383, 385, 388, 390, 391, 395, 404, 405, 414–419, 422, 423, 428, 429, 431, 433–435, 438–440, 443–445, 448, 450, 451, 455, 458, 460, 461, 464, 468–470, 476, 478, 479, 481–483, 486, 492, 493, 495, 497, 500, 509, 510, 515, 516, 520, 522, 525, 528, 530, 531, 533, 534, 536, 539–541, 545, 550, 556, 558, 564, 567, 570, 575, 576, 580, 582, 583, 589, 591, 597, 599, 601, 604, 609–611, 619, 620, 625, 649, 661, 675, 678, 689, 692, 693, 699, 715, 804, 806, 819, 899–907] [43, 53, 59, 61–63, 65–67, 72, 76, 77, 84,

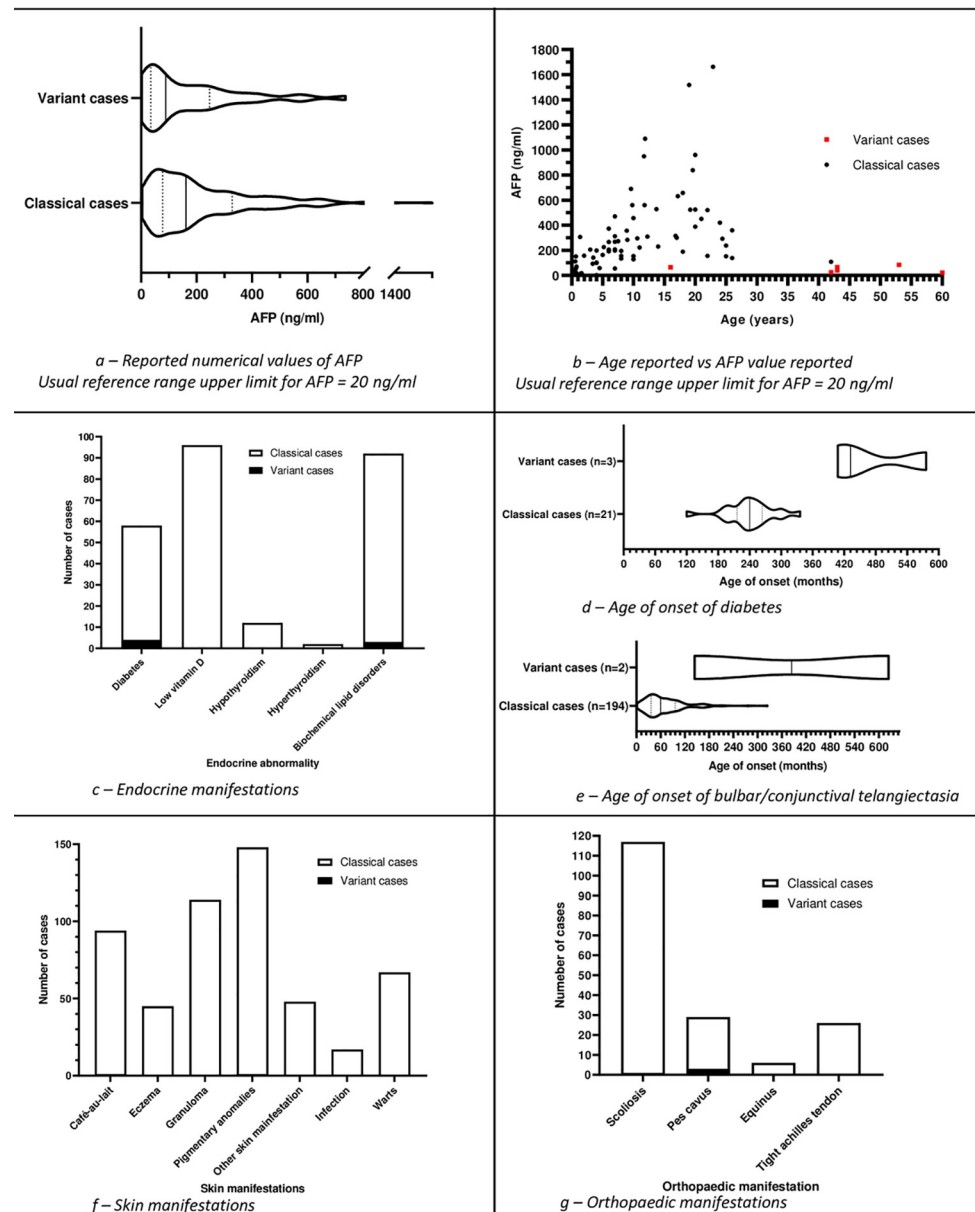

**Fig 7. Alpha fetoprotein (AFP), endocrine, bulbar telangiectasia, skin, and orthopaedic manifestations.**

92, 94–96, 102, 194, 203, 204, 207, 208, 211, 213, 214, 216–219, 221, 233, 362, 363, 368, 369, 375, 396, 398, 409, 411, 424, 481, 501, 504, 521, 532, 616, 628, 631, 641, 642, 768, 893]) reported raised levels of AFP. Reported individual AFP values and relationship between age and AFP level is reported in Fig 7A and 7B.

## Endocrine

Reported endocrine manifestations of A-T including diabetes, hypothyroidism and biochemical lipid disorders are described in Fig 7C and 7D.

In addition, 5 cases (1 study [75]) of rickets were reported.

### Dermatology

**Bulbar telangiectasia.** 2642 cases (346 studies [17–19, 23, 24, 29, 35–38, 42, 47, 49, 75, 80, 87, 91, 99–101, 103, 104, 106–113, 115, 116, 118–126, 128, 129, 131, 132, 134, 138, 139, 141, 143, 145, 147, 149, 150, 152, 155, 158,166, 168, 172, 174–176, 180, 181, 222–234, 236–240, 242, 244, 247, 248, 250, 251, 254–256, 258–263, 265–267, 278, 281–285, 287–290, 294, 298–300, 302, 304, 305, 307, 309, 310, 316, 317, 319, 323–326, 331, 332, 335, 338–340, 369, 372, 374–376, 382–385, 388–392, 401, 403–405, 407, 410–412, 414–418, 423–429, 431–433, 435, 439, 440, 444, 445, 447, 448, 450, 451, 455–457, 467–470, 472, 473, 476, 477, 513, 518, 520, 523–525, 527, 528, 530, 534–536, 543, 546, 547, 555, 556, 562, 563, 566, 569, 571–576, 580, 582, 583, 588, 589, 591–595, 598–604, 608–610, 651, 770, 841, 918–920] [47, 52, 53, 57, 58, 63, 66–68, 72, 76, 77, 79, 83, 84, 96, 183, 186, 189–194, 196, 197, 199, 202, 204–206, 211–213, 215, 217–221, 231, 269–276, 345–347, 349, 350, 353–355, 358, 361, 365, 366, 368, 378, 393–397, 406, 483, 485, 488, 489, 492–495, 497–499, 501–504, 521, 538, 540, 554, 611, 616, 618, 621, 626, 628, 630, 631, 633, 640, 646, 658, 718, 921–925]; reported bulbar of conjunctival telangiectasia. The age of presentation of bulbar or conjunctival telangiectasia is shown on Fig 7E.

294 cases (80 studies [18, 23, 24, 42, 52–54, 61, 66, 75, 76, 80, 84, 88, 91, 101, 104, 108, 111, 166, 175, 179, 190, 194, 212, 217, 221, 224, 225, 228, 230, 231, 236, 239, 242, 244, 248, 254, 259, 270, 271, 274, 289, 290, 315, 331, 347, 366, 372, 384, 394, 401, 403, 406, 414, 426, 428, 435, 448, 449, 457, 468, 483, 489, 492, 493, 498, 502, 518, 535, 547, 556, 573, 574, 592, 640, 651, 718, 841, 895]) reported other telangiectasia. A breakdown of these results is shown in S6 Fig. Other reported skin manifestations are shown in Fig 7F.

### Orthopaedics

Scoliosis, pes cavus abnormalities, equinus foot abnormalities and tight Achilles tendon(s) were reported as shown in Fig 7G.

Four cases reported age of diagnosis of scoliosis (median 131.4 months, range 102.0 months– 172.8 months). An additional 62 cases reported a mean age of diagnosis resulting in overall mean age of diagnosis (n = 66) of 153.0 months. One study [372] reported surgery for left thoracolumbar scoliosis at 14 years.

### Gastrointestinal

A variety of gastrointestinal manifestations and interventions were reported (Fig 8A and 8B).

The reported gastrostomy insertion indications are described in S7 Fig.

66 cases (14 studies [35, 226, 233, 345, 431, 448, 449, 463, 659, 743, 768, 818, 909, 916]) reported a diagnosis of fatty liver or hepatic steatosis and age of diagnosis was reported in 2 cases (252.0 months and 336.0 months). Seven cases were in the presumed/confirmed variant group and 59 cases were in the presumed/confirmed classical group.

### Other medical problems

The word cloud in S8 Fig shows other medical conditions that were reported in the literature that have not been reported elsewhere in this review.

### Reproductive health

7 studies [35, 52, 106, 119, 356, 396, 644] reported 12 cases of pregnancy (8 healthy infants in 4 cases and 8 further cases who were pregnant at least once). 6 presumed/confirmed classical cases and 6 presumed/confirmed variant cases. One study [644] reported one male who had 2 children. There were 2 case reports of primary [251, 636] and 2 cases of secondary [432, 636]

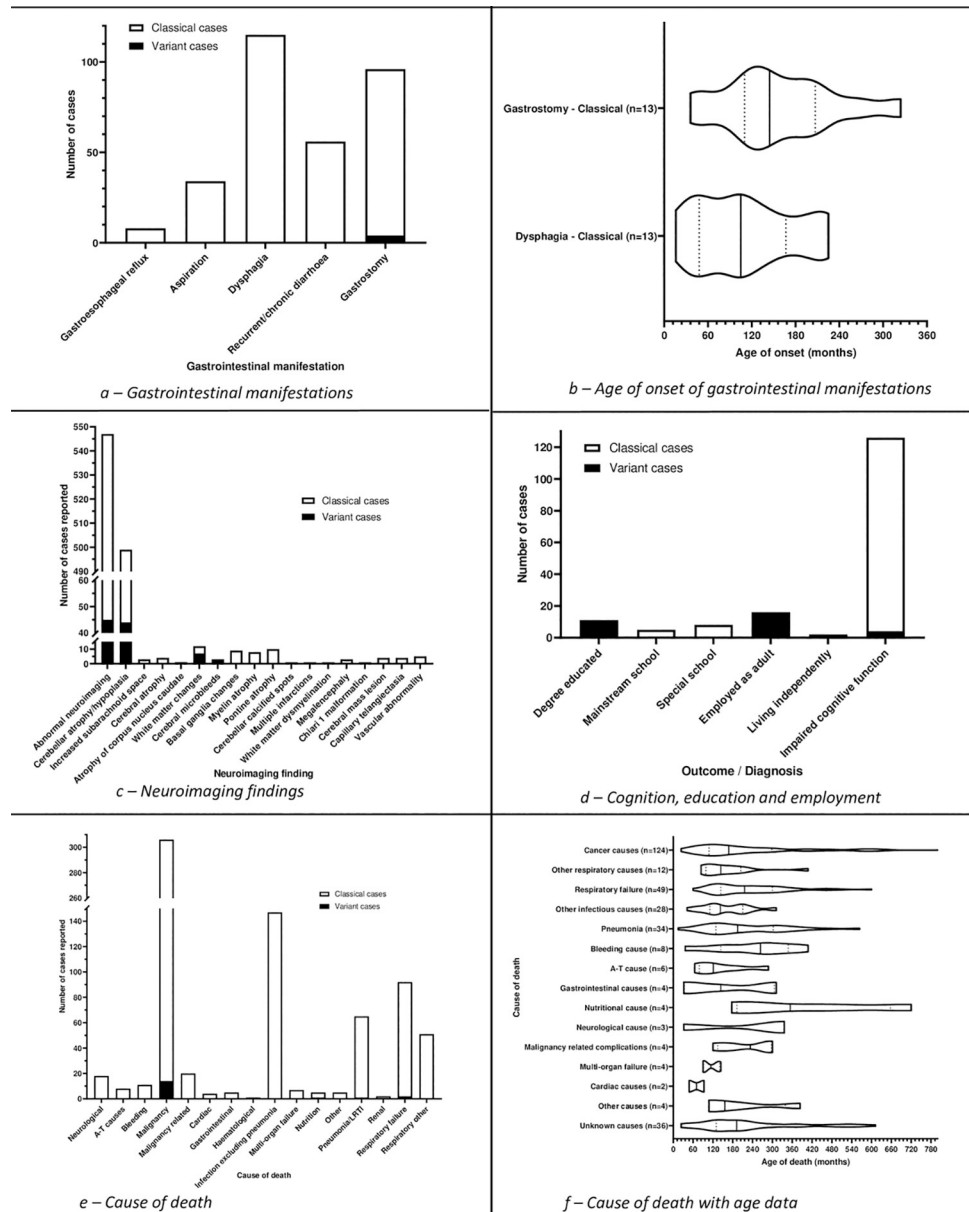

**Fig 8. Gastrointestinal, neuroimaging, cognitive and educational manifestations, and cause of death.**

amenorrhoea. 2 studies [111, 287] reported 7 cases of delayed menarche. 1 study [52] reported delayed sexual characteristics in 4 of 14 cases. One study [636] reported one case of no puberty by 19 years.

## Social outcomes

Included studies reported limited data on cognitive function, employment and education. The data that were reported are shown in Fig 8D.

As expected, there were several reports of delayed neurological development in early life (S9 Fig).

Table 4. Age of death in ataxia-telangiectasia.

| | Number of cases | Number of studies | | Age of death |
|---|---|---|---|---|
| | | | | Median (IQR, range) (months) |
| All cases | 294 | [160] | | 168.0 (120.0–279.0, 15.0–912.0) |
| | | [24, 28, 33, 35, 36, 43, 49, 52, 57, 61, 62, 70, 72, 75, 76, 78–80, 90–92, 99, 108, 114, 116, 121, 124, 128, 131, 133, 134, 139, 155, 160, 166, 177, 187, 190, 204, 220, 221, 225, 229, 231, 240, 242, 250, 255, 264, 275, 276, 278, 287, 290, 294, 296, 298, 304, 305, 310, 317, 319, 339, 345, 347, 350, 354, 364, 370, 373, 378, 382, 385, 391, 394, 397, 401, 407–410, 424, 427, 431, 432, 434, 438, 442, 445, 448, 449, 457, 458, 464, 467, 473, 474, 483, 489, 490, 495, 498, 538, 562, 563, 566, 567, 570, 574–576, 579, 603, 610, 614, 624, 626, 657, 669, 676, 681, 704, 711, 712, 718, 770, 771, 778, 781, 796, 798–800, 817, 818, 822, 825, 828, 831, 846, 851, 860, 861, 864, 871, 875, 888, 890, 891, 894, 896, 946–954] | | |
| Variant cases only | 18 | [12] | | 576.0 (420.0–612.0, 110.0–912.0 |
| | | [24, 28, 33, 35, 36, 43, 49, 52, 61, 62, 72] | | |
| Classical cases only | 277 | [147] | | 168.0 (108.0–259.5, 15.0–648.0) |
| | | [24, 36, 49, 52, 70, 75, 76, 78–80, 90–92, 99, 108, 114, 116, 121, 124, 128, 131, 133, 134, 139, 155, 160, 166, 177, 187, 190, 204, 220, 221, 225, 229, 231, 240, 242, 250, 255, 264, 275, 276, 278, 287, 290, 294, 296, 298, 304, 305, 310, 317, 319, 339, 345, 347, 350, 354, 364, 370, 373, 378, 382, 385, 391, 394, 397, 401, 407–410, 424, 427, 431, 432, 434, 438, 442, 445, 448, 449, 457, 458, 464, 467, 473, 474, 483, 489, 490, 495, 498, 538, 562, 563, 566, 567, 570, 574–576, 579, 603, 610, 614, 624, 626, 657, 669, 676, 681, 704, 711, 712, 770, 771, 778, 781, 796, 798–800, 817, 818, 822, 825, 828, 831, 846, 851, 860, 861, 864, 871, 875, 888, 890, 891, 894, 896, 946–954] | | |

## Death

1705 deaths were reported. 294 cases reported age of death (Table 4). 752 causes of death were reported in 687 cases. 1021 cases did not report a cause of death, or it was unknown (Fig 8E and 8F with further details in S6 Table in S1 File).

## Quality assessment of included studies

72 case control studies were quality assessed. The total number of stars (*) available was 10 with 10 stars representing the best quality. There were one, 10*; 6, 9*; 14, 8*; 17, 7*; 14, 6*; 11, 5*; 7, 4*; and one, 3* studies (see details in S7 Table in S1 File).

58 cohort studies were assessed. Using full criteria 56 studies were rated poor and 2 rated as fair when converted to AHRQ standards. Large numbers of downgrading were due to the lack of a control group. When this criterion was removed, of the 52 studies without a comparable group, 7 studies were rated poor, 29 fair, and 16 good. Details are given in S8 Table in S1 File. The 6 studies that had a comparable cohort were rated as 1 poor (abstract only), 1 fair (full text), and 4 good (all full text).

## Discussion

This review puts together a cohesive narrative of evidence based-information about A-T that will allow healthcare professionals and researchers to provide better information to families, and design and deliver research to improve care.

## Summary of evidence

We found a large volume of literature on A-T with over 1000 studies included in the analysis. Despite excluding duplicate cases, we found reports of 18247 cases. Most were classical A-T but 2.5% were reported as variants. The worldwide prevalence of variant A-T is not determined as yet.

This review contains cases of A-T from across the world with a large variety of phenotypic features in addition to the expected features including cerebellar gait ataxia and conjunctival/bulbar telangiectasia. There was a wide range in the age of cases reported.

Although cases were reported from 74 countries, nearly a quarter of the cases were from the USA and another quarter from just four other countries (the UK, Italy, Germany, and Turkey). The data presented may therefore be skewed towards presentations as seen in certain parts of the world. There are limited or no cases reported from several regions including Sub-Saharan Africa, parts of South America, and the Middle East. It is unlikely that A-T does not occur in these regions. This distribution may represent the global inequity in the care of children with A-T and a reporting/publication bias.

## Main findings

Although, as expected, most cases reported cerebellar ataxia, we found reports of cases with no cerebellar ataxia including 47 reports of classical A-T. These may be incomplete reports, inaccurate diagnoses, or could have been rarer presentations where other features such as leukaemia present before the ataxia manifests. Such reports, especially with a genetic diagnosis, are also more likely with screening pre-symptomatic young children such as when there is a family history.

In keeping with the existing view, we found that the median reported age of wheelchair requirement is 10 years. This requirement comes considerably later, by 26–27 years, in those with variant A-T. As expected, cerebellar gait ataxia was the most reported first presenting symptom however over a quarter did not have ataxia as their first clinical presentation. Dysarthria was reported as the first presentation in 9% of cases. Fewer reports of cases with typical presentations may be less likely to be published due to a bias towards reporting and publication of unusual presentations.

Although we found only a few cases, diagnosis in the newborn period due to screening of those with immunological abnormalities or family history is likely to become more common particularly following the introduction of routine screening for severe combined immunodeficiency disease in several countries including the UK. Such an early diagnosis may confer some benefit such as earlier provision of support for neurological signs and symptoms, treatment for related conditions such as bronchiectasis, and early diagnosis and management of malignancies.

As expected, median age of death was lower in classical cases (14 years 0 months) compared to variant cases (48 years 0 months), likely due to no ATM protein kinase activity resulting in a more severe phenotype in classical cases.

Raised AFP is often used as part of the diagnostic process. Although AFP results were reported in 158 studies, longitudinal results of AFP were very rarely presented. It was difficult to extract AFP data in relation to the time of diagnosis of the various clinical manifestations of A-T. Lower AFP at an older age was seen in those with variant A-T. A longitudinal study of AFP would help to show the pattern of AFP levels throughout the course of the disease and possibly lead to earlier diagnosis of malignancy, or clinical manifestations of A-T, enabling earlier treatments or supportive care.

## Secondary outcomes

As A-T is a rare disease, it is not unusual for the condition to be misdiagnosed. We found that, most often, A-T was mis-labelled as cerebral palsy (CP). Since delay in developmental milestones manifest first, the infant is labelled with CP before the recognition of ataxia. In addition, due to its rarity, and perhaps due to limited knowledge of the condition among physicians, A-T may not be considered initially. We found that classical cases were diagnosed at a median age of 6 years and variant cases at 29 years and 6 months. Variant A-T is often diagnosed much later in life when typical symptoms manifest, or a diagnosis is initially missed, or not considered, due to the milder phenotype.

Dystonia was a common feature in both variant and classical cases. Although data were limited to 43 cases, dystonia appears to present earlier in variant compared to classical cases. Dysarthria however was reported at a much older age in the variant group, compared to the classical group similar to oculomotor apraxia.

In comparison to the classical group, very few cases of recurrent infections were reported in variant cases, suggesting immunological impairment is not a common part of the variant phenotype. Despite interstitial lung disease being a recognised complication of A-T, only three cases reported the use of home oxygen. Bronchiectasis was reported more commonly than interstitial lung disease.

Lymphoma and leukaemia were the most common malignancies reported. Very few cases of lymphoma were reported in the variant group where we found reports of a wide variety of solid tumours. We are not aware of a routine screening protocol for malignancy in people with A-T in the UK, despite almost 10% of cases in this review reporting a history of at least 1 malignancy and there are likely to be many more that were not reported. However, some countries do have screening programmes for all people with A-T, which we think would be very helpful, by facilitating early diagnosis of malignancies.

Similarly, although difficulties with nutrition and swallowing are well known in A-T, we found very few cases, mostly of classical A-T, that reported gastrostomy insertion. Data were not sufficient to determine if gastrostomy insertion improved outcomes.

We found some cases of diabetes, youngest at the age of 10 years. Data were limited and we were unable to determine the presence of risk factor and types of treatment needed. There is growing evidence [961] for the development of hepatic steatosis/fatty liver and its association with A-T and we found 66 cases that reported hepatic steatosis and several that reported dyslipidaemia.

As expected, cerebellar atrophy was the most common neuropathological finding reported. Several studies reported mild, moderate or severe cerebellar atrophy, but none presented a standardised classification thus making it difficult to combine the reports. Limited data on EMG/nerve conduction studies were reported in the literature. Some reported peripheral neuropathy. Not much information was available about axonal neuropathy, particularly in children, however EMG is an uncomfortable procedure that is often not tolerated. Exploring this gap in our understanding may enable clinicians to diagnose unsafe swallowing or scoliosis earlier. A longitudinal EMG/nerve conduction study is needed.

Vitamin D deficiency is a concern in A-T exacerbated by advice to avoid sun exposure to reduce the risk of skin cancers. We found 152 cases that reported vitamin D levels and over a third were normal. This demonstrates that it is possible to maintain adequate vitamin D levels with supplementation and appropriate life-style advice.

We found reports of granulomatous disease only among classical cases suggesting that granulomas are linked to a lack of protein kinase. Similarly, scoliosis was only reported in classical cases suggesting that this is a feature of the more severe clinical phenotype.

Few studies reported IQ or cognitive function using a standardised and validated tool. We were unable to determine if A-T is associated with global impairment or if only specific domains are affected. Some cognitive tests are dependent on speech, motor movements and eye movements, and therefore it is difficult to test IQ in people with A-T demonstrating yet another gap in our knowledge of A-T.

## Strengths and limitations

Despite our comprehensive literature search and review, we did not find population-based studies and were unable to determine the prevalence of A-T. We have included a wide variety of studies to ensure a complete representation of the available literature. However, this made data extraction and synthesis a challenge. There is no standardised reporting format for A-T. Most case reports concentrate on positive findings and very few report the absence of signs or symptoms. Clinical features were, often, arbitrarily classified such as mild/moderate/severe and in the absence of a standardised classification, such reports could not be compared with each other.

We expect that, similar to other rare diseases, reports of A-T are subject to a reporting and publication bias. It is likely that rarer or unusual presentations are more likely to be published and the typical presentation may be under-represented in literature and, therefore, in this review. We also found several publications from same authors or the same centres. It is possible that some such reports will include the same cases. Duplicate reporting is also more likely in a condition such as A-T due to the multi-system involvement. The same case may be reported several times with publications focusing on a different aspect of the case each time. Where possible, we excluded identifiable duplicates, but it is likely that some may remain unnoticed. Due to the large volume of literature, it was unfeasible to contact authors and request further information on this or other matters. We were unable to access a few full text articles and were limited to English language reports.

We followed a standardised search strategy, data extraction, assessed quality of publications where possible, and combined the available data. Data were only extracted pre-intervention in interventional studies as the intervention could change the natural history of the disease. Where reports only presented non-specific information, data was excluded to ensure reliability. With attention to methodological rigour, we ensured that despite the limitations, this review is a concise yet exhaustive overview of A-T literature.

## Conclusion

A-T is a widely reported condition. We found that classical and variant cases are reported in many forms but there is a lack of standardised reporting and population-based studies. Well designed population-based longitudinal cohort studies are required to find the true prevalence and natural history of the condition. Development of core outcomes sets will further enable comparison between populations and cohorts if similar outcomes are reported in a standardised manner in all studies. Such epidemiological research will provide the high-quality evidence needed to improve care of those with A-T and their families and work towards trying to find a cure for this life-shortening disease.

## Supporting information

**S1 Checklist. PRISMA checklist.**
(PDF)

**S1 Protocol. Summary search protocol.**
(PDF)

**S1 File.** Table S1 Outcome definitions and, Table S2 Study type definitions and, Table S3 Reported antenatal problems and, Table S4 Reported birth weight and gestation and, Table S5 Reported postnatal problems and, Table S6 Detailed cause of death and, Table S7 Quality assessment Case-control studies and, Table S8 Quality assessment Cohort studies.
(PDF)

**S2 File. Fig and supplemental fig references.**
(PDF)

**S3 File. FINAL resubmission full dataset.**
(XLSX)

**S1 Fig. Incorrect, incomplete, and missed diagnoses.**
(TIF)

**S2 Fig. Breakdown of recurrent infections.**
(TIF)

**S3 Fig. Presenting symptoms of Hodgkin's Lymphoma.**
(TIF)

**S4 Fig. Presenting symptoms of non-Hodgkin's Lymphoma.**
(TIF)

**S5 Fig. Presenting symptoms of leukaemia.**
(TIF)

**S6 Fig. Other telangiectasia sites.**
(TIF)

**S7 Fig. Indication for gastrostomy.**
(TIF)

**S8 Fig. Other medical problems word cloud.**
(TIF)

**S9 Fig. Delayed neurological development in early life.**
(TIF)

## Author Contributions

**Conceptualization:** Emily Petley, Shalini Ojha, William P. Whitehouse.

**Data curation:** Emily Petley, Alexander Yule, Shalini Ojha, William P. Whitehouse.

**Formal analysis:** Emily Petley, Alexander Yule, Shaun Alexander, Shalini Ojha, William P. Whitehouse.

**Funding acquisition:** Emily Petley, William P. Whitehouse.

**Investigation:** Emily Petley, William P. Whitehouse.

**Methodology:** Emily Petley, Shalini Ojha, William P. Whitehouse.

**Project administration:** Emily Petley.

**Supervision:** Shalini Ojha, William P. Whitehouse.

**Writing – original draft:** Emily Petley.

**Writing – review & editing:** Emily Petley, Alexander Yule, Shaun Alexander, Shalini Ojha, William P. Whitehouse.

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
