## [Decision Letter · Decision Letter 0]

7 Jul 2021

PONE-D-21-14187

The Natural History of Ataxia-Telangiectasia (A-T): A Scoping Review

PLOS ONE

Dear Dr. Whitehouse,

Thank you for submitting your manuscript to PLOS ONE. After careful consideration, we
feel that it has merit but does not fully meet PLOS ONE’s publication criteria as it
currently stands. Therefore, we invite you to submit a revised version of the
manuscript that addresses the points raised during the review process.

Please submit your revised manuscript by Aug 14 2021 11:59PM. If you will need more
time than this to complete your revisions, please reply to this message or contact
the journal office at plosone@plos.org. When
you're ready to submit your revision, log on to https://www.editorialmanager.com/pone/ and select the 'Submissions
Needing Revision' folder to locate your manuscript file.

If you would like to make changes to your financial disclosure, please include your
updated statement in your cover letter. Guidelines for resubmitting your figure
files are available below the reviewer comments at the end of this letter.

We look forward to receiving your revised manuscript.

Kind regards,

Tai-Heng Chen, M.D.

Academic Editor

PLOS ONE

Journal Requirements:

2. Please confirm that you have included all items recommended in the PRISMA
checklist including:

- the full electronic search strategy used to identify studies with all search terms
and limits for at least one database.

- Update search and analysis to include studies published since December 2019

4. We note that Figures 3 and 4 in your submission contains map images which may be
copyrighted. All PLOS content is published under the Creative Commons Attribution
License (CC BY 4.0), which means that the manuscript, images, and Supporting
Information files will be freely available online, and any third party is permitted
to access, download, copy, distribute, and use these materials in any way, even
commercially, with proper attribution. For these reasons, we cannot publish
previously copyrighted maps or satellite images created using proprietary data, such
as Google software (Google Maps, Street View, and Earth). For more information, see
our copyright guidelines: http://journals.plos.org/plosone/s/licenses-and-copyright.

1. You may seek permission from the original copyright holder of Figures 3 and 4 to
publish the content specifically under the CC BY 4.0 license.  

5. We note that this manuscript is a systematic review or meta-analysis; our author
guidelines therefore require that you use PRISMA guidance to help improve reporting
quality of this type of study. Please upload copies of the completed PRISMA
checklist as Supporting Information with a file name “PRISMA checklist".

Reviewers' comments:

Reviewer's Responses to Questions

**Comments to the Author**

1. Is the manuscript technically sound, and do the data support the conclusions?

Reviewer #2: Partly

Reviewer #3: Yes

Reviewer #4: Partly

2. Has the statistical analysis been performed
appropriately and rigorously? 

Reviewer #2: Yes

Reviewer #3: N/A

Reviewer #4: I Don't Know

3. Have the authors made all data underlying the
findings in their manuscript fully available?

Reviewer #2: Yes

Reviewer #3: Yes

Reviewer #4: Yes

4. Is the manuscript presented in an intelligible
fashion and written in standard English?

Reviewer #2: No

Reviewer #3: Yes

Reviewer #4: No

5. Review Comments to the Author

Reviewer #2: Authors present an extensive review on the Natural History of
Ataxia-Telangiectasia (A-T) by critically compiling all human studies that reported
any aspect of of A-T.They confirm the multi-system involvement in A-T, neurological
symptoms being the most frequent presenting features in classical A-T. They also
illustrate manifestations of variant forms of the disease.

The study is well written, describes clearly results. The methods used to retrieve
studies from scientific databases and their critical review leading to the selection
of 1163 full text articles are fairly described. The introduction brings the
necessary elements for understanding of the field. Discussion is well conducted and
contains a paragraph on “limitations of the study” which is appreciated.

Although not bringing very new findings, the study collected retrospectively clinical
data from from more than 1000 reported patients, allowing a reliable quantification
of the key clinical markers the disease, making of the study an interesting
reference for clinicians.

I have a few important comments that should be addressed:

Results section:

First paragraph: 167 The search yielded 14622 titles and abstracts (Figure 1). After
removal of 180268 duplicates and exclusion of 13459 articles by review of title and
abstract, 1163 full text articles were reviewed. Results do not sum up in paragraph
one of results, in particular in the first sentence. Has to be checked

References sources are missing on too many occasions in the text. This aspect is of
course crucial in he context of a review article and requires a thorough
reassessment.

One example Line 223: Error! Reference source not found; line

Authors confirm the increased risk of malignancies. For example: “ lymphoma and
leukaemia were the most common malignancies reported”. More information regarding
mean age of onset should be described in the main body of the text, as an important
clinical information element. Age of onset is mentioned as a secondary endpoint in
the methods, so this information should be available.

The paragraph on EMG: EMG description is limited to normal vs abnormal criteria which
is very vague. Authors should try to provide more detailed information on sensory
nerve conduction vs CMAP, velocities etc…

Discussion Line 427:”As A-T is a rare disease, it is not unusual for the condition to
be misdiagnosed. We found that, most often, A-T was mis-labelled as cerebral palsy
(CP)”. There are several forms of CP. Do authors refer to the Ataxic form? If yes
this should be added.

Minor comments / edits/ typos:

There are still too many edit typing errors in the text, that should would require a
careful review.

A few examples of typing errors:

-Table 3 typos: “ataxia-telangiectasia”

- Line 411: “ataxia-telangiectasia”

Reviewer #3: Dear authors, congratulations to this extensive review you performed and
the very comprehensive overview you provide within the text, tables and figures. I
recommend to accept the publication with minor revision.

Overall comment: I would suggest to not entitle this work a scoping review but much
more a systematic review. The authors conducted a structured and extensive
literature search and extraction of full texts as well as data in a systematic way.
Moreover, the rated each study where applicable by the recommended risk of bias
tools. You should then include a paragraph with the information on the level of
evidence of the included publications rated by one of the established LoE tools.

Methods: The authors describe the search terms. I would like to suggest to also
include a paragraph on the initial PICO question that informed the literature
search.

Discussion:

Line 466 Vit D levels were normal in a third of reported data sets - due to
supplementation or without any supplementation? Should VitD be supplemented in any
case?

I would like to suggest to give a short outlook on overall management in terms of a
multimodal interdisciplinary approach. You may include an outlook on future novel
therapeutic approaches, if applicable (i.e. any kind of molecular/genetic therapy in
the pipeline?).

References:

Some references are not correctly embedded. Please check troughout the manuscript and
reference list.

Reviewer #4: The intention was good and such a review is certainly needed. The
authors searched through a huge amount of papers (1163 full text articles).

Primary and secondary outcomes were defined.

I have included minor points in the attached document.

As for the major points, I would certainly recommend that phenotype - genotype
correlation is included in the paper. In line with this, authors should define
"classical" and "variant" phenotype based on the type of the mutations. Or at least,
a comparison should be provided. It is not sufficient to state that all the cases
without detailed information were claswsified as "classical".

6. PLOS authors have the option to publish the peer
review history of their article (what does this mean?). If published, this will
include your full peer review and any attached files.

If you choose “no”, your identity will remain anonymous but your review may still be
made public.

**Do you want your identity to be public for this peer review?** For
information about this choice, including consent withdrawal, please see our
Privacy Policy.

Reviewer #2: **Yes: **Nicolas Deconinck,Pediatric neurology department;
Hôpital universitaire des Enfants reine Fabiola, Université Libre de Bruxelles,
Belgium

Reviewer #3: No

Reviewer #4: No

---

## [Author Response · Author response to Decision Letter 0]

11 Nov 2021

We have revised the submission in line with the reviewers constructive comments, and
attach detailed responses with this submission.

to editor v3 31-10-2021.docx
---

## [Decision Letter · Decision Letter 1]

27 Dec 2021

PONE-D-21-14187R1The Natural History of
Ataxia-Telangiectasia (A-T): A Systematic ReviewPLOS
ONE

Dear Dr. Whitehouse,

Thank you for submitting your manuscript to PLOS ONE. After careful consideration, we
feel that it has merit but does not fully meet PLOS ONE’s publication criteria as it
currently stands. Therefore, we invite you to submit a revised version of the
manuscript that addresses the points raised during the review
process.

Please submit your revised manuscript by Feb 10 2022 11:59PM. If you will need more
time than this to complete your revisions, please reply to this message or contact
the journal office at plosone@plos.org. When
you're ready to submit your revision, log on to https://www.editorialmanager.com/pone/ and select the 'Submissions
Needing Revision' folder to locate your manuscript file.

Please include the following items when submitting your revised
manuscript:A rebuttal letter that responds to each point raised by the academic
editor and reviewer(s). You should upload this letter as a separate file
labeled 'Response to Reviewers'.A marked-up copy of your manuscript that highlights changes made to the
original version. You should upload this as a separate file labeled
'Revised Manuscript with Track Changes'.An unmarked version of your revised paper without tracked changes. You
should upload this as a separate file labeled 'Manuscript'.

If you would like to make changes to your financial disclosure, please include your
updated statement in your cover letter. Guidelines for resubmitting your figure
files are available below the reviewer comments at the end of this letter.

We look forward to receiving your revised manuscript.

Kind regards,

Tai-Heng Chen, M.D.

Academic Editor

PLOS ONE

Journal Requirements:

Reviewers' comments:

Reviewer's Responses to Questions

**Comments to the Author**

1. If the authors have adequately addressed your comments raised in a previous round
of review and you feel that this manuscript is now acceptable for publication, you
may indicate that here to bypass the “Comments to the Author” section, enter your
conflict of interest statement in the “Confidential to Editor” section, and submit
your "Accept" recommendation.

Reviewer #5: All comments have been addressed

Reviewer #6: (No Response)

2. Is the manuscript technically sound, and do the data
support the conclusions?

Reviewer #5: Yes

Reviewer #6: Yes

3. Has the statistical analysis been performed
appropriately and rigorously? 

Reviewer #5: Yes

Reviewer #6: Yes

4. Have the authors made all data underlying the
findings in their manuscript fully available?

Reviewer #5: Yes

Reviewer #6: Yes

5. Is the manuscript presented in an intelligible
fashion and written in standard English?

Reviewer #5: Yes

Reviewer #6: Yes

6. Review Comments to the Author

Reviewer #5: This paper is a systematic review for a rare disease - ataxia
telangiectasia. The authors searched and reviewed the literature extensively, and
provided useful information in presenting the whole picture of this disease. I
believe this paper makes contribution to the existing literature and is worth
publishing.

I only have a few minor comments.

1. Line 218: “The median delay in diagnosis (n=17) was 0.0 i.e., diagnosed at first
presentation but there was wide variation with a range of 0.0-312.0 (IQR, 0.0-43.0)
months.” This sentence is hard to understand. Please edit it.

2. The subtitle “Cerebellar signs” in Line 264 partially overlaps with the subtitle
“Ataxia and Mobility” in Line 230. Please consider integrating these two
sections.

3. Line 430: in Discussion, the authors said “In keeping with the existing view, we
found that the median reported age of wheelchair requirement is 10 years.” But I
cannot find this statement in Results.

Reviewer #6: The authors carefully review 1234 full-text articles to the natural
history of ataxia-telangiectasia (A-T); however, the natural history cannot be
strongly supported with limited evidence of predominantly case reports or case
series. Indeed, it would be hard to do population-based study to fill the evidence
gap. Overall, the article is well written and I believe it would provide useful
information for the clinicians and the scientists.

7. PLOS authors have the option to publish the peer
review history of their article (what does this mean?). If published, this will
include your full peer review and any attached files.

If you choose “no”, your identity will remain anonymous but your review may still be
made public.

**Do you want your identity to be public for this peer review?** For
information about this choice, including consent withdrawal, please see our
Privacy Policy.

Reviewer #5: **Yes: **Tzu-Pu Chang

Reviewer #6: **Yes: **Yang-Pei Chang

---

## [Author Response · Author response to Decision Letter 1]

26 Jan 2022

Thank you for your further review. We have considered carefully the 3 minor comments
made:

1. “Line 218: “The median delay in diagnosis (n=17) was 0.0 i.e., diagnosed at first
presentation but there was wide variation with a range of 0.0-312.0 (IQR, 0.0-43.0)
months.” This sentence is hard to understand. Please edit it.”

We have edited this and tried to clarify it. We added: "Most reported cases were
diagnosed without any delay, however a minority were diagnosed late: the median
delay in diagnosis (n=17) was 0.0 i.e., diagnosed at first presentation but there
was wide variation with a range of 0.0-312.0 (IQR, 0.0-43.0) months.” This is in the
Manuscript and in the track changes version resubmitted.

2. “The subtitle “Cerebellar signs” in Line 264 partially overlaps with the subtitle
“Ataxia and Mobility” in Line 230. Please consider integrating these two
sections.”

We are grateful to Tzu-Pu Chang for their comments, and we agree that there is
overlap between the "Cerebellar signs" and "Ataxia & Mobility". However, we feel
it important not to assume that the mobility problems or even ataxia is always
cerebellar ataxia, as children with A-T can have ataxia and mobility impairments
caused by frequent myoclonus, chorea, and tremors, as well as neuropathic weakness.
So, we believe it is better to report them separately, based on the terminology used
in the papers we reviewed.

3. “Line 430: in Discussion, the authors said “In keeping with the existing view, we
found that the median reported age of wheelchair requirement is 10 years.” But I
cannot find this statement in the results.” 

Figure 4b shows all reported age data for cerebellar gait ataxia, truncal ataxia,
limb ataxia and mobility, including “Wheelchair-bound Classical (n=109)” showing the
median age for onset 120 months (10 years).

We hope that the clarifications are helpful.

Systematic review Resub response to reviewer
26-01-2022.docx
---

## [Decision Letter · Decision Letter 2]

7 Feb 2022

The Natural History of Ataxia-Telangiectasia (A-T): A Systematic Review

PONE-D-21-14187R2

Dear Dr. Whitehouse,

We’re pleased to inform you that your manuscript has been judged scientifically
suitable for publication and will be formally accepted for publication once it meets
all outstanding technical requirements.

Kind regards,

Tai-Heng Chen, M.D.

Academic Editor

PLOS ONE

Reviewers' comments:

Reviewer's Responses to Questions

**Comments to the Author**

1. If the authors have adequately addressed your comments raised in a previous round
of review and you feel that this manuscript is now acceptable for publication, you
may indicate that here to bypass the “Comments to the Author” section, enter your
conflict of interest statement in the “Confidential to Editor” section, and submit
your "Accept" recommendation.

Reviewer #5: All comments have been addressed

2. Is the manuscript technically sound, and do the data
support the conclusions?

Reviewer #5: Yes

3. Has the statistical analysis been performed
appropriately and rigorously? 

Reviewer #5: Yes

4. Have the authors made all data underlying the
findings in their manuscript fully available?

Reviewer #5: Yes

5. Is the manuscript presented in an intelligible
fashion and written in standard English?

Reviewer #5: Yes

6. Review Comments to the Author

Reviewer #5: The authors have addressed all my comments. In my opinion, this paper
contributes to the literature in this field and is worth publishing.

7. PLOS authors have the option to publish the peer
review history of their article (what does this mean?). If published, this will
include your full peer review and any attached files.

If you choose “no”, your identity will remain anonymous but your review may still be
made public.

**Do you want your identity to be public for this peer review?** For
information about this choice, including consent withdrawal, please see our
Privacy Policy.

Reviewer #5: **Yes: **Tzu-Pu Chang

---

## [Editor Report · Acceptance letter]

15 Feb 2022

PONE-D-21-14187R2 

The Natural History of Ataxia-Telangiectasia (A-T): A Systematic Review 

Dear Dr. Whitehouse:

I'm pleased to inform you that your manuscript has been deemed suitable for
publication in PLOS ONE. Congratulations! Your manuscript is now with our production
department. 

Kind regards, 

on behalf of

Dr. Tai-Heng Chen 

Academic Editor

PLOS ONE